# Application of Radiomics in Melanoma: A Systematic Review and Meta-Analysis

**DOI:** 10.3390/cancers17193130

**Published:** 2025-09-26

**Authors:** Rosa Falcone, Sofia Verkhovskaia, Francesca Romana Di Pietro, Chiara Scianni, Giulia Poti, Maria Francesca Morelli, Paolo Marchetti, Federica De Galitiis, Matteo Sammarra, Armando Ugo Cavallo

**Affiliations:** 1Department of Oncology, Istituto Dermopatico dell’Immacolata IDI-IRCCS, 00167 Rome, Italy; r.falcone@idi.it (R.F.); s.verkhovskaia@idi.it (S.V.); g.poti@idi.it (G.P.); m.morelli@idi.it (M.F.M.); p.marchetti@idi.it (P.M.); f.degalitiis@idi.it (F.D.G.); 2Department of Radiology, Istituto Dermopatico dell’Immacolata IDI-IRCCS, 00167 Rome, Italy; c.scianni@idi.it (C.S.); m.sammarra@idi.it (M.S.); a.cavallo@idi.it (A.U.C.)

**Keywords:** melanoma, radiomics, immunotherapy, targeted therapy, predictive, meta-analysis

## Abstract

Radiomics is being increasingly investigated as a tool to support clinical decisions in different solid tumors. The aim of our review was to analyze the state of the art of radiomic applications in melanoma. We found that metastatic disease and immunotherapy were the settings most explored through computed tomography imaging and three-dimensional analysis. The prediction of treatment response was the most investigated outcome. Radiomic models can achieve strong discriminative performance with low to moderate heterogeneity in methodologies. Limited validation strategies need to be overcome through greater standardization and transparency, and further verified in large prospective cohorts.

## 1. Introduction

Solid tumors show a wide spatial and temporal heterogeneity. Some diagnostic methods, such as tissue biopsies, analyzing small fragments of the solid mass, allow us to identify the cancer histotype at the expense of losing dynamic biological differences. In contrast, medical imaging offers a non-invasive way to examine the entire tumor, capturing its complexity in a complete and repeatable manner. Thanks to the progress in imaging technology, contrast agents and standardized protocols, it is now possible to extract detailed, quantitative data from scans. This has led to the development of radiomics, a technique that uses computer algorithms to extract advanced features from medical images, such as shape, intensity, texture and spatial relationships, which can reflect the underlying biological characteristics of tumors and offer valuable insights into tumor behavior [1]. Radiomics has the potential to identify image-based biomarkers that reflect the tumor’s phenotype and microenvironment, helping clinicians make better decisions regarding diagnosis, prognosis and treatment [2].

Melanoma is an aggressive skin cancer, driven by both environmental and genetic factors. Ultraviolet radiation induces DNA damage and mutations, contributing to genomic instability, while infrared radiation, although non-mutagenic, may promote melanoma by inhibiting apoptosis in damaged melanocytes. Among the main genes implicated in genetic susceptibility are MC1R, CDKN2A, TP53 and components of the MAPK pathway, which regulate pigmentation, DNA repair and cell proliferation [3]. Standard treatments include immune checkpoint inhibitors (ICIs), such as anti-programmed death (PD)-1 antibodies, anti-cytotoxic T-lymphocyte antigen (CTLA)-4 antibodies, and, for patients with BRAF V600 mutations, targeted therapies with BRAF and MEK inhibitors [4]. In the context of this malignancy, radiomics represents a non-invasive, reproducible diagnostic tool that may help us to predict responses to therapy or survival outcomes, provide prognostic significance, be used to assess tumor heterogeneity, be used to detect mutational status, and even be used to evaluate the presence of tumor-infiltrating lymphocytes (TILs), especially CD8+ T cells, supporting more personalized treatment approaches [5]. Most studies extract radiomic features from Positron Emission Tomography (PET)/computed tomography (CT) imaging with 18F-fluorodeoxyglucose (FDG), CT imaging, Magnetic Resonance Imaging (MRI) or ultrasound [6,7,8,9]. In a large dataset involving 503 lesions from 71 patients with metastatic melanoma treated with ICIs, machine learning models using CT radiomic features predicted treatment responses and survival with high accuracy (Area Under the Curve, AUC, 0.88–0.91), demonstrating the potential of radiomics across different patients and clinical settings [10]. Another study found that texture skewness, a measure of asymmetry in image intensity, was associated with progression-free survival (PFS) and overall survival (OS) in melanoma patients treated with pembrolizumab [11]. More recent studies using MRI-based radiomic models have shown improved accuracy, reaching up to 83% in detecting BRAF mutations in brain metastases from melanoma [12]. Radiomics has also been used to explore tumor immune profiles. For example, tumors with high mean positive pixel (MPP) values on CT scans were associated with worse survival outcomes, and with lower levels of CD8+ T cells, suggesting an “immune-cold” phenotype, which tends to respond poorly to ICIs [13].

These findings suggest that radiomics could serve as a valuable, non-invasive method for discovering imaging biomarkers and developing tailored therapeutic strategies. However, to be applied reliably in clinical settings, radiomics needs validation through larger, multicenter studies with harmonized imaging and analysis workflows. Moreover, it should be integrated with laboratory and molecular data.

Radiomic feature extraction is typically performed using software tools such as PyRadiomics, LifeX and TexRAD, which allow the analysis of both two-dimensional (2D) and three-dimensional (3D) images [14,15,16]. A major limitation in the field is the lack of standardization. Variations in imaging protocols, scanner types, tumor segmentation methods and feature extraction techniques can introduce inconsistency and reduce reproducibility [17,18].

With the aim of synthesizing the existing literature and quantitatively evaluating the effectiveness of radiomic models in predicting therapy responses in melanoma, we conducted a systematic review and meta-analysis.

## 2. Materials and Methods

### 2.1. Search Strategy and Study Selection

This systematic review was conducted according to the PRISMA 2020 guidelines. A comprehensive search of the literature was performed on 1 March 2025, using three major electronic databases: PubMed, Scopus and Web of Science (WOS). The goal was to identify original research articles applying radiomics to the management of melanoma, regardless of imaging modality, disease stage or clinical setting.

A PICO (Population, Intervention, Comparison, Outcome) question was used to define the key elements of the research [19]. Our Population included patients with melanoma; the Intervention consisted of the radiomic analysis of medical imaging (e.g., CT, MRI, PET, ultrasound); a Comparison group was not included, as the objective of this review is to systematically evaluate the current applications and outcomes of radiomic analysis in melanoma, rather than to compare it with other diagnostic or prognostic methods; the Outcomes were diagnostic accuracy, prognostic stratification, and treatment response prediction.

The detailed search string was: “(“textural” OR “radiomics” OR “texture” OR “histogram”) AND (“melanoma”) AND (“computed tomography” OR “CT” OR “magnetic resonance” OR “MRI” OR “MR” OR “PET” OR “Positron Emission Tomography” OR “Ultrasound” OR “US”)”.

The search yielded a total of 331 records (PubMed: 112; Scopus: 169; WOS: 50). After removing 26 duplicates, 305 unique records were screened based on titles and abstracts.

Two review authors (R.F. and S.V.) independently and manually screened the results of the search strategies for eligibility for this review by reading the abstracts, with disagreements resolved by consensus. Of those 305 abstracts, 263 were excluded due to irrelevance or not meeting the inclusion criteria (Figure 1). The full text of 42 articles was sought for further assessment. Two reviewers (A.U.C. and C.S.) independently performed the assessment, with disagreements resolved by consensus.

Two reports were excluded during full-text review (one for being written in a non-English language and one for not involving radiomics). Ultimately, 40 studies were included in the qualitative synthesis (Appendix A) [10,11,20,21,22,23,24,25,26,27,28,29,30,31,32,33,34,35,36,37,38,39,40,41,42,43,44,45,46,47,48,49,50,51,52,53,54,55,56].

To increase the accuracy, repeatability and completeness of our analysis, the PRISMA 2020 checklist was completed and is available as Appendix A. The risk of bias of the included studies was assessed using the QUADAS-2 tool [57]. Two reviewers (A.U.C. and R.F.) independently performed the assessment, with disagreements resolved by consensus.

For the diagnostic meta-analysis, two readers (A.U.C. and R.F.) screened manuscripts to identify studies that evaluated the prediction of the response to therapy. Disagreements were resolved by consensus. 

The certainty of the evidence for each outcome included in the meta-analysis (AUC and sensitivity) was evaluated using the GRADE approach [58], adopted for diagnostic test accuracy studies. No sensitivity analyses were performed due to the limited number of included studies and substantial heterogeneity. In addition to extracting measures of discrimination (AUC), we also assessed whether the included studies reported calibration performance or decision curve analysis.

The protocol of the systematic review and meta-analysis was not prospectively registered.

### 2.2. Eligibility Criteria and Data Extraction

Eligible studies included patients with melanoma, employing radiomic analysis of imaging data, and reporting at least one clinical endpoint. Only original, full-text, peer-reviewed articles in English were considered.

The extracted data were the number of patients and lesions; imaging modality and software; type of analysis (2D or 3D); type of therapy; presence of molecular data; clinical setting (advanced vs. localized); and primary endpoint. The endpoints were categorized into five groups: therapy response, survival prediction, mutational status, differential diagnosis and technical analysis.

### 2.3. Statistical Analysis

Statistical analysis was performed with R V 4.4.2 [R Core Team (2021). R: A language and environment for statistical computing. R Foundation for Statistical Computing, Vienna, Austria. URL https://www.R-project.org/)].

Given the high percentage of studies in the categories “therapy response” and “survival prediction” (24 out of 40, 60%), we decided to assess the eligibility of the studies to be included in a diagnostic meta-analysis.

The diagnostic meta-analysis was conducted using the random-effects model to synthesize diagnostic performance across multiple studies based on reported AUC values. In manuscripts where a 95% confidence interval was reported instead of the standard error, the standard error was calculated using the following formula:(CI upper limit − CI lower limit)/(2 × 1.96).

A random-effects model was fitted with Restricted Maximum Likelihood (REML) estimation. The potential publication bias was evaluated with Egger’s test and using a funnel plot.

## 3. Results

### 3.1. Study Selection

The PRISMA flow diagram summarizes the selection process (Figure 1). After deduplication and screening, 40 studies met the inclusion criteria and were analyzed in full (Appendix A). The most common reasons for exclusion included the absence of radiomic analysis, insufficient methodological details and irrelevant populations or outcomes.

### 3.2. Descriptive Analysis of Included Studies

The 40 selected studies provide a broad overview of radiomic applications in melanoma. A clear increase in the number of publications emerged over time. Only nine studies (22.5%) were published prior to 2020, indicating that radiomics in melanoma was still a relatively niche topic in earlier years. From 2020 onward, there has been a notable surge in research activity: seven studies (17.5%) were published in 2020, followed by four (10.0%) in 2021. The trend peaked in 2022, with nine studies (22.5%), and remained high in 2023, with eight studies (20.0%). The years 2024 and 2025 showed a slight decline, with two (5.0%) and one study (2.5%), respectively (Figure 2).

A total of 4673 patients and 24,561 lesions were included in the analysis. The vast majority of studies (34 out of 40, 85%) included patients with advanced melanoma, while only 3 studies focused on localized disease and 3 did not report the stage. Immunotherapy was the most frequent treatment reported, appearing in 20 studies (50%). A few studies involved targeted therapy (three studies, 7.5%), or unspecified therapies or indiscriminately ICI and target therapy (five studies, 12.5%). In 12 studies (30%), the treatment type was either unclear or not applicable to the radiomic model.

CT, including dual-energy CT, was the most commonly used imaging technique, employed in 17 studies (42.5%), followed by MRI in 7 studies (17.5%) and PET/CT (20%). Multimodal combinations, such as CT plus MRI or PET/CT, were used in several cases (15%). Rare imaging sources exploited ultrasound (5%).

The most frequent clinical endpoint was the prediction of therapy response, assessed in 17 studies (42.5%). In 5 of these 17 studies, a double endpoint was examined: therapy response and survival prediction. Differential diagnosis, mutational status and technical analysis were evaluated in five (12.5%), eight (20%) and three (7.5%) studies, respectively. Survival prediction was the only endpoint in 17.5% of the papers.

Studies were heterogeneous in endpoints within the same category. Indeed, some explored clinical benefit [25], others the response by RECIST1.1 or PERCIT [21], and others hyperprogression [28] or tumoral heterogeneity or pseudoprogression [20]. With regard to survival, the prediction of overall survival or progression-free survival at different time points (6 and 12 months) were the most commonly used endpoints [21,24].

The prediction of molecular alterations most often involved the analysis of the BRAF or NRAS gene. Some studies developed a model using radiomic features to predict the BRAF mutational status [12,26] in patients with melanoma brain and lung metastases. The diagnostic performance was encouraging in patients with brain metastases using MRI; on the contrary, it was unsuccessful in lung metastasis prediction using CT scan images. Two papers investigated the correlation between biological and image biomarkers: a positive correlation with CD8 expression was found [33], and a radiomic signature was developed to predict the circulating tumor DNA allele fraction [35].

Differential diagnosis was performed between benign pigmented lesions and melanoma, or comparing different types of malignant cancers in hidden organs (squamous esophageal carcinoma versus melanoma of the same district; different mass of the eye).

In terms of technical methodology, 3D radiomic analysis was used in 29 studies (72.5%), while 2D analysis was reported in 7 studies (17.5%). The dimensionality was not specified in four studies.

In the QUADAS-2 assessment (Appendix A), the Index Test domain showed a high risk of bias in 30 studies. For the Reference Standard, nine studies were rated as “unclear” and three as “high risk,” while for Flow and Timing, nine were “unclear” and one “high risk,” with the same numbers observed for applicability concerns.

### 3.3. Meta-Analysis

Out of the 24 studies focusing on therapy response and survival prediction, 9 [21,23,24,25,27,28,29,30,34] provided the necessary statistical data (AUC, SE and/or CI) for inclusion in the meta-analysis (Table 1). The remaining 15 studies were excluded for the following reasons: (1) six studies lacked a reported measure of uncertainty (SE or CI); (2) nine studies were focused on survival analysis or other topics and did not report an AUC. One study [34] provided two analyses on two different populations, which were included separately. Thus, ultimately, 10 radiomic models were included in the final assessment. In cases where a study reported multiple radiomic models, the one with the highest AUC was selected for inclusion in the meta-analysis. This decision was made to ensure consistency across studies and to reflect the best-performing predictive capacity reported by each author.

Overall, 1989 patients and 606 lesions were included in the final meta-analysis.

The pooled estimate of the AUC was 0.83 (95% CI: 0.74 to 0.92), indicating strong overall discriminatory ability of the models included (Figure 3). The heterogeneity across studies was low to moderate, with an I^2^ of 28.6%, τ^2^ = 0.0049, and the Q-test for heterogeneity was not statistically significant (Q = 8.61, *p* = 0.4741). No evidence of publication bias was found based on the regression test for funnel plot asymmetry (z = −0.72, *p* = 0.470) (Figure 4).

In accordance with GRADE guidance, we started from high certainty because well-conducted observational accuracy studies begin at this level. We downgraded two levels for risk of bias because 7/9 studies were at high risk in the index test domain: radiomic models were frequently evaluated using internal cross-validation with non-independent data, and index test interpretation was not blinded to the reference standard, both likely to inflate accuracy estimates. We downgraded one additional level for imprecision because the pooled estimate showed a wide 95% CI (0.74–0.92) that spans clinically relevant thresholds. We did not downgrade for indirectness, as the populations, index tests (radiomics on CT/MRI/PET) and outcomes aligned with our review question. The overall GRADE score for the body of evidence was rated as very low.

### 3.4. Subgroup Analyses

Because some studies reported outcomes at the patient level whereas others reported outcomes at the lesion level, we performed subgroup analyses to account for this methodological difference. Studies reporting patient-level outcomes (6 studies, 1989 patients) demonstrated a pooled and predicted AUC of 0.89 (95% CI: 0.82–0.96; prediction interval: 0.82–0.96), whereas studies reporting lesion-level outcomes (4 studies, 606 lesions) showed a lower pooled and predicted AUC of 0.74 (95% CI: 0.62–0.86; prediction interval: 0.62–0.86). Both subgroups exhibited no heterogeneity (I^2^ = 0%, τ^2^ = 0), and the Q-test for heterogeneity was not statistically significant (Q = 3.03, *p* = 0.69, for patient-level studies and Q = 1.01, *p* = 0.78, for lesion-level studies). Egger’s tests were non-significant (*p* = 0.16 and *p* = 0.30).

The meta-analysis with pooled studies (patient- and lesion-level) yielded a lower pooled and predicted AUC of 0.83 (95% CI: 0.74–0.92; prediction interval: 0.74–0.92) with moderate heterogeneity (I^2^ = 28.6%); the Q-test for heterogeneity was not statistically significant (Q = 8.61, *p* = 0.4741). This estimate lies between the higher performance of patient-level studies (AUC 0.89; I^2^ = 0%) and the lower performance of lesion-level studies (AUC 0.74; I^2^ = 0%). Publication bias was not detected (Egger’s test *p* = 0.47). These findings indicate that methodological differences in the unit of analysis (patient vs. lesion) contribute to the variation in reported accuracy, and that combining the two approaches results in an intermediate overall effect. The data from the patient- and lesion-level analysis are reported in Table 2 and Figure 5.

Additionally, we performed subgroup analyses based on the imaging modality (CT and PET-CT) and validation strategy (internal validation and cross-validation) (Table 3 and Appendix A). In the subgroup analyses by imaging modality, CT-based studies (six studies) demonstrated a pooled and predicted AUC of 0.87 (95% CI: 0.77–0.98; prediction interval: 0.77–0.98) with low heterogeneity (I^2^ = 7.6%, τ^2^ = 0.0019); the Q-test for heterogeneity was not statistically significant (Q = 3, *p* = 0.7). PET-based studies (four studies) showed a lower pooled and predicted AUC of 0.81 (95% CI: 0.67–0.95; prediction interval: 0.67–0.95) with moderate heterogeneity (I^2^ = 39.9%, τ^2^ = 0.0078); the Q-test for heterogeneity was not statistically significant (Q = 3.99, *p* = 0.26).

When stratified by validation approach, studies using cross-validation (six studies) yielded a pooled and predicted AUC of 0.79 (95% CI: 0.67–0.90; prediction interval: 0.67–0.90) with moderate heterogeneity (I^2^ = 27.7%, τ^2^ = 0.005); the Q-test for heterogeneity was not statistically significant (Q = 4.84, *p* = 0.43). Studies employing internal validation (two studies) achieved a higher pooled and predicted AUC of 0.92 (95% CI: 0.82–1.01; prediction interval: 0.82–1.01) with no heterogeneity (I^2^ = 0%, τ^2^ = 0); the Q-test for heterogeneity was not statistically significant (Q = 0.001, *p* = 0.89). One study by Dittrich D. et al. [34] reported two separated analyses without validation and was not included in the subgroup analysis. Egger’s tests did not suggest publication bias in any subgroup (all *p* > 0.05), except for the internal validation subgroup, where the test could not be computed due to the limited number of studies (two studies).

Subgroup analyses by endpoint were not performed, as the endpoints were highly heterogeneous, and further subdivision was considered likely to produce misleading or non-informative results.

## 4. Discussion

This systematic review and meta-analysis aims to provide an updated and comprehensive overview of how radiomics is currently applied in the field of melanoma, with particular focus on its ability to predict treatment response. The results highlight both the potential and the current limitations of this rapidly evolving area of research. Our work is by far the largest and most recent systematic review on this topic. So far, only one systematic review has been published and dates back to 2020 [59]. It was limited to the inclusion of 10 articles compared to the 40 of this analysis. The distribution of the published papers over time confirms a growing and sustained interest in radiomics as a tool for improving melanoma care, particularly in the last five years, which together account for over 75% of all the included studies.

Among the 40 studies included in this review, the majority (85%) focused on patients with advanced melanoma, and immunotherapy was the most frequently studied treatment (50%). This reflects the clinical relevance and growing use of ICIs in routine practice, given its versatility and broad applicability in both early-stage and advanced disease settings. Most studies (40%) employed CT-based imaging, as it is the cornerstone imaging modality for melanoma assessment, whilst MRI and hybrid PET/CT scans were less frequently used. This may impact the generalizability of the results. Interestingly, 3D analysis was much more common than 2D, reinforcing the notion that volumetric approaches are becoming standard in radiomic workflows.

Radiomics has been applied across several domains, from diagnosis to staging, prognosis, and treatment and response assessment—in particular, in the monitoring of tumor response (by RECIST or PERCIST); the prediction of survival outcomes, especially with ICIs; tumor characterization predicting the subtype of cancer; and the genomic correlation with BRAF mutations. Therapy response was the most frequent endpoint, addressed in 60% of the included studies. However, even within this category, we observed substantial variation in how the response was defined and measured. Some studies considered overall response rates, others focused on organ-specific responses (e.g., hepatic metastases) or interorgan heterogeneity, and others aimed to predict phenomena such as hyperprogression [20]. This conceptual heterogeneity is one of the key challenges in radiomic research, as it limits the comparability and reproducibility of findings. In some cases, a multimodality approach of blood markers and radiomic features was established [36] to predict pseudoprogression or hyperprogression [28]. Survival status was another predicted outcome based on pretreatment 18F-FDG PET radiomic features [30] or CT features [29] in patients treated with ICIs for melanoma.

Although used less frequently, radiomic approaches were used to predict BRAF mutational status, in a non-invasive manner, both in brain and lung metastases [12,26,32,38], exploiting MRI for brain metastases and CT for lung lesions. The results showed the approaches to be feasible and promising for prediction based on MRI images of brain metastases, whilst they were negative for lung metastases. Detecting BRAF V600 status based on 18F-FDG PET/CT alone did not provide clinically relevant knowledge [39].

One study focused on the possibility of predicting the response to targeted treatment specifically in BRAF-mutated patients, showing the difficulty in the radiological interpretation of the response in this subtype of patients [40].

Another potential application of radiomics consists of differential diagnosis between benign lesions (naevi) and melanoma [31] by ultra-high-frequency ultrasound or different histotypes of malignant tumors in hidden organs, such as the esophagus or eye [22,37]. The authors attempt to differentiate squamous esophageal cancer from mucosal melanoma of the esophagus and uveal melanoma from other intraocular masses, respectively. Recently, Teng-Li Lin at al. developed a new approach, based on machine learning algorithms, called the Spectrum-Aided Vision Enhancer (SAVE) to enhance the visualization of skin lesions and increase the accuracy of classification (cancerous versus non-cancerous tissues), achieving 98% accuracy [60]. This approach has good potential for real-world clinical use where timely diagnosis is crucial, if confirmed across more diverse sets of skin images and less expensive equipment. Interestingly, the applications of radiomics go beyond cutaneous melanoma and also include uveal and mucosal melanoma [23].

The advantages of radiomics are multiple: it is not invasive, is quantitative and is objective, making it potentially repeatable and cost-effective (using imaging acquired for standard care).

A meta-analysis was conducted on the subset of studies that reported both the AUC and SE for models predicting therapy response. The analysis, which included 9 papers and 10 models, found a pooled AUC of 0.83 (95% CI: 0.74 to 0.92). The prediction interval ranged from 0.669 to 0.997, suggesting that future studies might show similarly high performance with some variability. The heterogeneity across studies was low to moderate (I^2^ of 28.6%), and no evidence of publication bias was found.

However, the results of the meta-analysis must be interpreted with caution due to a number of important methodological limitations. First, in studies reporting multiple models, we selected the one with the highest AUC for inclusion in the meta-analysis. While this choice aligns with common practice and reflects the best-case scenario, it may introduce optimism bias. Additionally, there is variability between the purpose of the studies and endpoints used for predictions, which may be a source of heterogeneity. Although all the included studies fall under the broad category of “therapy response” and “survival prediction”, the operational definitions varied widely, from organ-specific prediction (e.g., only hepatic metastases) to different time points of response (3, 6, 12 months), and distinct outcomes such as clinical benefit or hyperprogression. This variability complicates the interpretability of the pooled estimate. Moreover, six out of nine studies relied exclusively on cross-validation, without an independent test set, and one study with two models provided no validation [34]. Only two studies performed internal validation, which is a more robust method for estimating generalizability.

These results, while encouraging—with favorable pooled AUC values and low heterogeneity—are nonetheless biased by the limited number of studies on the topic and the overall scarcity of literature in this area. The lack of calibration assessment and clinical utility analyses represents an important limitation, as AUC alone does not fully reflect model performance or net clinical benefit. Future studies should incorporate calibration methods and decision curve analysis to provide a more comprehensive evaluation of prediction models.

The results from QUADAS-2 and GRADE assessment showed a high risk of bias in the systematic review and low quality of evidence in the meta-analysis, highlighting that the current research methodology in this field is generally of poor quality. There is an urgent need for further, high-quality studies to strengthen the evidence base.

Our analysis has several limitations. First, although the literature search was comprehensive, it was limited to studies published in English, and two potentially relevant full texts could not be retrieved. Second, the meta-analysis included only nine studies, which limits the robustness and generalizability of its conclusions. Most of the included studies focus on CT imaging; few ones use PET-CT scans. Therefore, the results are less generalizable to PET. Moreover, MRI and ultrasound are not considered at all. Another limitation is that four included studies that reported outcomes at the lesion level rather than at the patient level. Such reporting may introduce clustering effects, as multiple lesions from the same patient are not statistically independent, potentially leading to an underestimation of variance and an overstatement of precision. Because the necessary lesion-level counts were not provided, we were unable to perform hierarchical adjustment or recalculate standard errors. To address this, we conducted subgroup analysis comparing lesion-level and patient-level studies. Nonetheless, the lack of hierarchical adjustment should be considered a potential source of bias.

Finally, the diversity of radiomic pipelines, from image acquisition to feature selection and model validation, makes direct comparisons between studies difficult and highlights the need for greater standardization in the field.

## 5. Conclusions

Radiomics represents a promising avenue for improving the prediction of therapy response in melanoma. The available evidence suggests that these models can achieve strong discriminative performance, especially when applied to CT-based imaging in advanced disease. However, the analysis shows a high risk of bias and low quality of evidence, with limited validation strategies and inconsistent reporting practices, currently hindering the translation of radiomic models into clinical practice.

Future studies should prioritize methodological rigor, including the use of independent validation cohorts, clearly defined clinical endpoints, and adherence to reporting standards. Only through greater standardization and transparency radiomics will it be able to fulfill its potential as a clinically valuable tool in the management of melanoma.

## Figures and Tables

**Figure 1 cancers-17-03130-f001:**
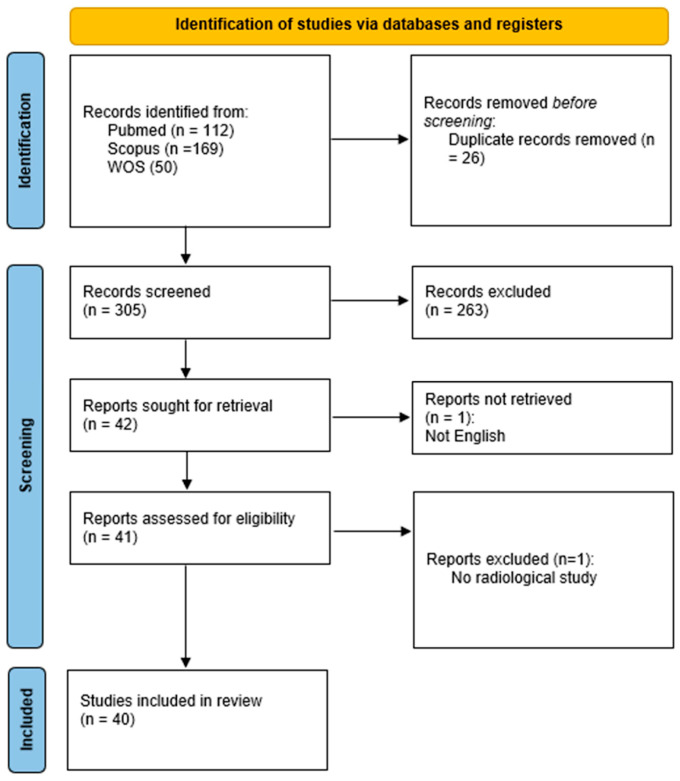
PRISMA flow chart.

**Figure 2 cancers-17-03130-f002:**
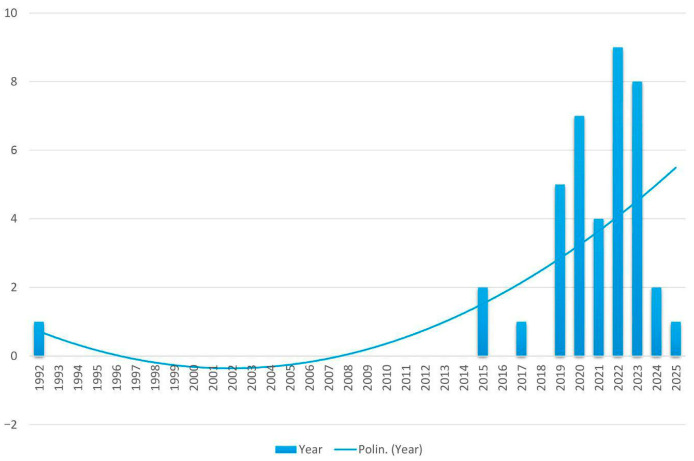
Trend over time of studies on radiomics in melanoma.

**Figure 3 cancers-17-03130-f003:**
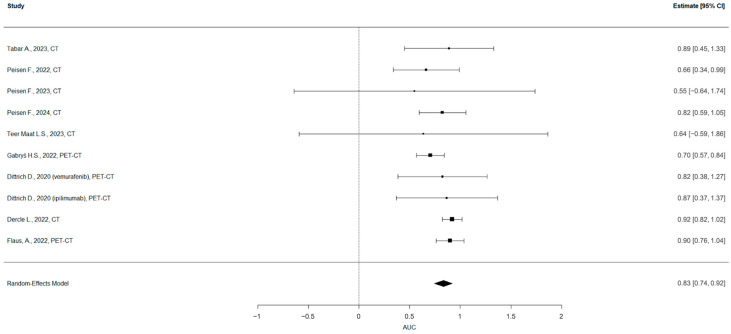
Forest plot of the diagnostic meta-analysis. The pooled estimate of the AUC was 0.83 (95% CI: 0.74 to 0.92).

**Figure 4 cancers-17-03130-f004:**
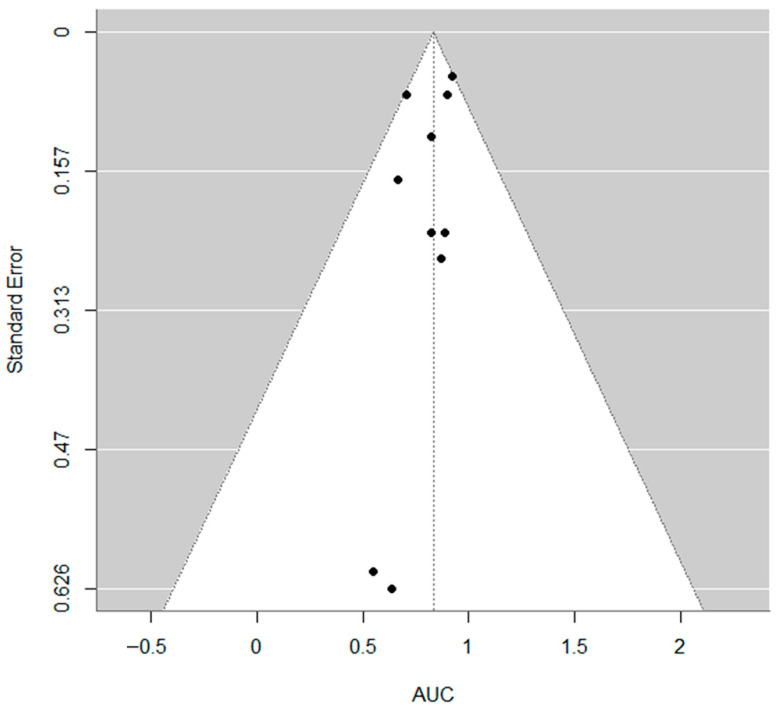
Funnel plot. No evidence of publication bias was found based on the regression test for funnel plot asymmetry (z = −0.72, *p* = 0.470).

**Figure 5 cancers-17-03130-f005:**
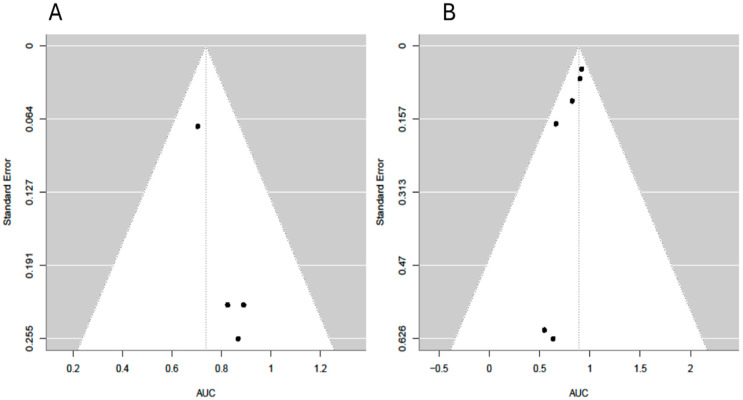
Funnel plot of subgroup analysis of lesion-level studies (**A**) and patient-level studies (**B**).

**Table 1 cancers-17-03130-t001:** Summary of studies included in meta-analysis.

Study	Imaging	N° pt.	Validation	Endpoint
Tabari A., 2023; [23] 10.3390/cancers15102700	CT	79	Internal validation	Prediction of hepatic mets response (3 mo)
Peisen F., 2022; [27] 10.3390/cancers14122992	CT	262	Cross-validation	Prediction of response (3 mo); OS (6, 12 mo)
Peisen F., 2023; [24] 10.3390/diagnostics13203210	CT	91	Cross-validation	Prediction of BOR; PFS (6 mo); OS (6, 12 mo)
Peisen F., 2024; [21] 10.3390/cancers17010001	CT	146	Cross-validation	Prediction of BOR; PFS (6, 9, 12 mo); OS (6 mo)
Ter Maat L.S., 2023; [25] 10.1016/j.ejca.2023.02.017	CT	620	Cross-validation	Prediction of clinical benefit for a minimum of 6 mo
Gabryś H.S., 2022; [28] 10.3389/fonc.2022.977822	PET-CT	56	Cross-validation	Hyperprogression (3 mo)
Dittrich D., 2020; [34] 10.1055/a-1140-5458	PET-CT	9	No validation	Response (3 mo) according to PERCIST
Dittrich D., 2020; [34] 10.1055/a-1140-5458	PET-CT	17	No validation	Response (3 mo) according to PERCIST
Dercle L., 2022; [29] 10.1001/jamaoncol.2021.6818	CT	575	Internal validation	Prediction of response (3 mo); OS (6 mo)
Flaus A., 2022; [30] 10.3390/diagnostics12020388	PET-CT	56	Cross-validation	Prediction of OS, PFS (12 mo)

CT: computed tomography; PET: positron emission tomography; N° pt: number of patients; mo: months; mets: metastases; BOR: best overall response; PFS: progression-free survival; OS: overall survival; PERCIST: PET Response Evaluation Criteria in Solid Tumors.

**Table 2 cancers-17-03130-t002:** Subgroup analysis at patient and lesion level.

Subgroup	Number of Studies	I^2^	Pooled AUC	Predicted AUC	Egger’s Test *p* Value
Patient level	6	0.00	0.89 (95% CI: 0.82–0.96)	0.89 (95% CI: 0.82–0.96)	0.16
Lesion level	4	0.00	0.74 (95% CI: 0.62–0.86)	0.74 (95% CI: 0.62–0.86)	0.30
Pooled studies	10	28.6%	0.83 (95% CI: 0.74–0.92)	0.83 (95% CI: 0.74–0.92)	0.47

**Table 3 cancers-17-03130-t003:** Subgroup analyses of studies included in meta-analysis.

Subgroup	N° of Studies	I^2^	Pooled AUC	Predicted AUC	Egger’s Test *p* Value
CT	6	7.62	0.87 (95% CI: 0.77–0.98)	0.87 (95% CI: 0.77–0.98)	0.20
PET-CT	4	39.91	0.81 (95% CI: 0.67–0.95)	0.81 (95% CI: 0.67–0.95)	0.83
Cross-validation	6	27.66	0.79 (95% CI: 0.67–0.9)	0.79 (95% CI: 0.67–0.9)	0.53
Internal validation	2	0.00	0.92 (95% CI: 0.82–1.01)	0.92 (95% CI: 0.82–1.01)	NA

## Data Availability

Data are available in the main text and Appendix A.

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
