# Peer review of "Application of Radiomics in Melanoma: A Systematic Review and Meta-Analysis"

_cancers, 2025, doi:10.3390/cancers17193130_

Round 1
Reviewer 1 Report
Comments and Suggestions for Authors
No description of the risk of bias framework exists in prediction model investigations. Consider the evaluation rates utilizing PROBAST-AI for ML/AI prediction models and QUADAS-AI for diagnostic accuracy studies; additionally, it may be pertinent to report CLAIM items to address AI-specific reporting in imaging.
AUC and SE were published in only 9 out of 24 response-prediction studies, which exacerbates availability bias and undermines the efficacy of funnel-plot and Egger tests. Methods exist to generate or estimate C-statistics and standard errors based on previously reported data to mitigate bias from selective inclusion; please clarify why these methods were not utilized, or apply them and re-estimate. One should not overlook pooling on an appropriately modified scale when feasible.
The study demonstrates non-inferiority to the pooled AUC (0.83) of diverse endpoints, therapies, timepoints, and validation types (internal versus external). I request subgroup or meta-regression analyses categorized by endpoint, therapeutic kind, imaging modality, validation type, and feature type. Additionally, clustering may compromise precision when aggregating lesion-level count measurements without appropriate hierarchical adjustment. This stratification is advised in guidance documents regarding the synthesis of evidence for prediction models.
The majority of the included research, along with the meta-analysis, concentrated on AUC. Determine if control studies have provided calibration and clinical utility where relevant; if absent, document this as a restriction, as AUC alone does not inherently indicate net benefit. Another consideration is a summary of whether a study has implemented DCA and whether it should be undertaken in future research.
The predominance of CT (42.5%) may impact generalizability. Kindly provide commentary on the applicability of findings to FDG-PET/CT and MRI, highlighting the presence of several melanoma immunotherapy radiomics research and meta-analytic reviews, and address the modality-specific syntheses.
It is recommended to articulate the clinical necessity and the significance of this review more explicitly, using contemporary sources pertinent to the research such as:
1) Lin, Teng-Li, Arvind Mukundan, Riya Karmakar, Praveen Avala, Wen-Yen Chang, and Hsiang-Chen Wang. "Hyperspectral Imaging for Enhanced Skin Cancer Classification Using Machine Learning." Bioengineering 12, no. 7 (2025): 755.
Author Response
Thanks for the chance to review the manuscript and the insightful comments.
- No description of the risk of bias framework exists in prediction model investigations. Consider the evaluation rates utilizing PROBAST-AI for ML/AI prediction models and QUADAS-AI for diagnostic accuracy studies; additionally, it may be pertinent to report CLAIM items to address AI-specific reporting in imaging.
We thank the reviewer for this suggestion. In this study we applied QUADAS-2, as our primary aim was to synthesize and evaluate the results reported in the included studies, rather than to systematically assess the methodological quality of radiomics research.
While tools such as Radiomics Quality Score (RQS) or METRICS (Methodological Radiomics Score) could indeed be applied for that purpose, a comprehensive evaluation of study quality falls beyond the scope of the present work and will be the focus of a dedicated forthcoming manuscript.
- AUC and SE were published in only 9 out of 24 response-prediction studies, which exacerbates availability bias and undermines the efficacy of funnel-plot and Egger tests. Methods exist to generate or estimate C-statistics and standard errors based on previously reported data to mitigate bias from selective inclusion; please clarify why these methods were not utilized, or apply them and re-estimate. One should not overlook pooling on an appropriately modified scale when feasible.
We thank the reviewer for this insightful comment. As stated in the manuscript, we made every effort to retrieve missing data, including a thorough review of supplementary materials. However, in most cases AUC confidence intervals or standard errors were not available, and the information reported did not allow reliable back-calculation of standard error. Applying imputation methods under these conditions would have required unverifiable assumptions and, in our view, would have introduced additional bias. For this reason, we chose to transparently report the limitation rather than apply potentially misleading estimates.
- The study demonstrates non-inferiority to the pooled AUC (0.83) of diverse endpoints, therapies, timepoints, and validation types (internal versus external). I request subgroup or meta-regression analyses categorized by endpoint, therapeutic kind, imaging modality, validation type, and feature type. Additionally, clustering may compromise precision when aggregating lesion-level count measurements without appropriate hierarchical adjustment. This stratification is advised in guidance documents regarding the synthesis of evidence for prediction models.
We thank the reviewer for highlighting this important issue.
Four of the included studies reported lesion-level results, which may artificially inflate precision if clustering within patients is not accounted for. As lesion-level counts required for hierarchical adjustment were not reported in manuscripts, recalculation of standard errors was not feasible.
To address this, we performed subgroup analyses separating studies reporting patient-level (n=6) versus lesion-level outcomes (n=4).
Patient-level studies demonstrated a pooled and predicted AUC of 0.89 (95% CI: 0.82–0.96; prediction interval: 0.82–0.96), while lesion-level studies yielded a lower pooled and predicted AUC of 0.74 (95% CI: 0.62–0.86; prediction interval: 0.62–0.86). Both subgroups showed no heterogeneity (I²=0%) and Egger’s tests were non-significant (p=0.16 and p=0.30).
These findings indicate that lesion-level reporting is associated with systematically lower performance, and we have highlighted this in the revised manuscript.
In addition, we have acknowledged in the revised manuscript that the lack of hierarchical adjustment may bias variance estimates, and have emphasized this as a limitation.
We added these paragraphs in the results section:
Because some studies reported outcomes at the patient level whereas others reported outcomes at the lesion level, we performed subgroup analyses to account for this methodological difference. Studies reporting patient-level outcomes (6.1989 patients) demonstrated a pooled and predicted AUC of 0.89 (95% CI: 0.82–0.96; prediction interval: 0.82–0.96), whereas studies reporting lesion-level outcomes (4.606 lesions) showed a lower pooled and predicted AUC of 0.74 (95% CI: 0.62–0.86; prediction interval: 0.62–0.86). Both subgroups exhibited no heterogeneity (I²=0%), and Egger’s tests were non-significant (p=0.16 and p=0.30).
The meta-analysis with pooled studies (patient- and lesion-level) yielded a lower pooled and predicted AUC of 0.83 (95% CI: 0.74–0.92; prediction interval: 0.74–0.92) with moderate heterogeneity (I²=28.6%). This estimate lies between the higher performance of patient-level studies (AUC 0.89; I²=0%) and the lower performance of lesion-level studies (AUC 0.74; I²=0%). Publication bias was not detected (Egger’s test p=0.47). These findings indicate that methodological differences in the unit of analysis (patient vs. lesion) contribute to variation in reported accuracy, and that combining the two approaches results in an intermediate overall effect.
|
Subgroup |
Number of studies |
I2 |
Pooled AUC |
Predicted AUC |
Egger test p value |
|
Patient level |
6 |
0.00 |
0.89 (95% CI: 0.82–0.96) |
0.89 (95% CI: 0.82–0.96) |
0.16 |
|
Lesion level |
4 |
0.00 |
0.74 (95% CI: 0.62–0.86) |
0.74 (95% CI: 0.62–0.86) |
0.30 |
|
Pooled studies |
10 |
28.6% |
0.83 (95% CI: 0.74–0.92) |
0.83 (95% CI: 0.74–0.92) |
0.47 |
Additionally, we performed subgroup analyses based on imaging modality (CT and PET-CT) and validation strategy (internal validation and cross validation).
In subgroup analyses by imaging modality, CT-based studies (6) demonstrated a pooled and predicted AUC of 0.87 (95% CI: 0.77–0.98; prediction interval: 0.77–0.98) with low heterogeneity (I²=7.6%), whereas PET-based studies (4) showed a lower pooled and predicted AUC of 0.81 (95% CI: 0.67–0.95; prediction interval: 0.67–0.95) with moderate heterogeneity (I²=39.9%).
When stratified by validation approach, studies using cross-validation (6) yielded a pooled and predicted AUC of 0.79 (95% CI: 0.67–0.90; prediction interval: 0.67–0.90) with moderate heterogeneity (I²=27.7%), while those employing internal validation (2) achieved a higher pooled and predicted AUC of 0.92 (95% CI: 0.82–1.01; prediction interval: 0.82–1.01) with no heterogeneity (I²=0%).
One study by Dittrich D. et al reported two separated analyses without validation and was not included in the subgroup analysis.
Egger’s tests did not suggest publication bias in any subgroup (all p>0.05), except for the internal validation subgroup, where the test could not be computed due to the limited number of studies (k=2).
Subgroup analyses by endpoint were not performed, as endpoints were highly heterogeneous, and further subdivision was considered likely to produce misleading or non-informative results.
|
Subgroup |
Number of studies |
I2 |
Pooled AUC |
Predicted AUC |
Egger test p value |
|
CT |
6 |
7.62 |
0.87 (95% CI: 0.77–0.98) |
0.87 (95% CI: 0.77–0.98) |
0.20 |
|
PET |
4 |
39.91 |
0.81 (95% CI: 0.67–0.95) |
0.81 (95% CI: 0.67–0.95)
|
0.83 |
|
Cross validation |
6 |
27.66 |
0.79 (95% CI: 0.67–0.9)
|
0.79 (95% CI: 0.67–0.9) |
0.53 |
|
Internal validation |
2 |
0.00 |
0.92 (95% CI: 0.82–1.01) |
0.92 (95% CI: 0.82–1.01) |
NA |
Paragraph in Limitations:
Another limitation is that four included studies reported outcomes at the lesion level rather than at the patient level. Such reporting may introduce clustering effects, as multiple lesions from the same patient are not statistically independent, potentially leading to an underestimation of variance and an overstatement of precision. Because the necessary lesion-level counts were not provided, we were unable to perform hierarchical adjustment or recalculate standard errors. To address this, we conducted subgroup analysis comparing lesion-level and patient-level studies. Nonetheless, the lack of hierarchical adjustment should be considered a potential source of bias.
- The majority of the included research, along with the meta-analysis, concentrated on AUC. Determine if control studies have provided calibration and clinical utility where relevant; if absent, document this as a restriction, as AUC alone does not inherently indicate net benefit. Another consideration is a summary of whether a study has implemented DCA and whether it should be undertaken in future research.
We thank the reviewer for this important comment. Following the suggestion, we reviewed all included studies for information on calibration and clinical utility. None of the studies reported decision curve analysis (DCA), and only one study (Teer Maat L.S., 2023) presented a calibration curve. We have now added this information in the Methods, Results, and Discussion sections of the revised manuscript and emphasized the lack of calibration and clinical utility assessment as a relevant limitation and an important avenue for future research.
Section Materials and Methods: In addition to extracting measures of discrimination (AUC), we also assessed whether included studies reported calibration performance or decision curve analysis (DCA).
Section Results: Among the included studies, none reported DCA, and only one study (Teer Maat L.S., 2023) presented a calibration curve.
Discussion: The lack of calibration assessment and clinical utility analyses represents an important limitation, as AUC alone does not fully reflect model performance or net clinical benefit. Future studies should incorporate calibration methods and decision curve analysis to provide a more comprehensive evaluation of prediction models.
- The predominance of CT (42.5%) may impact generalizability. Kindly provide commentary on the applicability of findings to FDG-PET/CT and MRI, highlighting the presence of several melanoma immunotherapy radiomics research and meta-analytic reviews, and address the modality-specific syntheses.
Thanks for this comment.
The majority of research studies on radiomics rely on the use of CT, as it is the cornerstone imaging modality for melanoma assessment, as documented by previous research (10.3390/diagnostics13193065). Recently, the updated version of ESMO guidelines for melanoma staging recommend CT or PET-CT for neck, thorax, abdomen, pelvis plus brain MRI (10.1016/j.annonc.2024.11.006). In regard to therapy, immunotherapy is the treatment of choice in melanoma research, given its versatility and broad applicability in both early-stage and advanced disease settings.
We modified the discussion, integrating this comment:
Among the 40 studies included in this review, the majority (85%) focused on patients with advanced melanoma, and immunotherapy was the most frequently studied treatment (50%). This reflects the clinical relevance and growing use of ICIs in routine practice, given its versatility and broad applicability in both early-stage and advanced disease settings. Most studies (40%) employed CT-based imaging, as it is the cornerstone imaging modality for melanoma assessment, whilst MRI and hybrid PET-CT scans were less frequently used. This may impact generalizability of the results. Interestingly, 3D analysis was much more common than 2D, reinforcing the notion that volumetric approaches are becoming standard in radiomic workflows.
- It is recommended to articulate the clinical necessity and the significance of this review more explicitly, using contemporary sources pertinent to the research such as: 1) Lin, Teng-Li, Arvind Mukundan, Riya Karmakar, Praveen Avala, Wen-Yen Chang, and Hsiang-Chen Wang. "Hyperspectral Imaging for Enhanced Skin Cancer Classification Using Machine Learning." Bioengineering 12, no. 7 (2025): 755.
Thanks for this suggestion. We added the reference.
Recently, Teng-Li Lin at al. developed a new approach, based on machine learning algorithms, called the Spectrum-Aided Vision Enhancer (SAVE) to enhance the visualization of skin lesions and increase the accuracy of classification (cancerous versus non-cancerous tissues), achieving 98% (10.3390/bioengineering12070755). This approach has good potential for real-world clinical use where timely diagnosis is crucial, if confirmed across more diverse sets of skin images and less expensive equipment. […]
The advantages of radiomics are multiple: it is not invasive, quantitative and objective, therefore potentially repeatable, cost-effective (using imaging acquired for standard care).

Reviewer 2 Report
Comments and Suggestions for Authors
In the manuscript entitled “Application of radiomics in melanoma: a systematic review and meta-analysis” the authors present a comprehensive systematic review and meta-analysis of the current applications of radiomics in melanoma.
This manuscript is valuable, however some issues should be addressed before publication.
Strengthen and contextualize the etiology of melanoma in the introduction (refer to recent literature DOI: 10.3390/cancers17111784) by providing a solid context to highlight the relevance of radiomics as a non-invasive investigative tool.
Revise the final sentence on line 93. The concluding sentence of the introduction, "With the aim to study the current applications and the state of art of radiomics in melanoma, we carried out a systemic review and meta-analysis," is somewhat generic. Since your abstract specifically mentions the meta-analysis on therapy response prediction, the introduction should foreshadow this key aspect. A more specific reformulation would be: "With the aim of synthesizing the existing literature and quantitatively evaluating the effectiveness of radiomic models..., we conducted a systematic review and meta-analysis..."
Improve Figure 2. The current graph showing the trend over time is not visually impactful. For a clearer and more intuitive representation of discrete counts per year, replacing the line graph with a bar chart (histogram) would more effectively illustrate the number of studies published annually.
Author Response
Thanks for the chance to integrate and enrich the manuscript.
- Strengthen and contextualize the etiology of melanoma in the introduction (refer to recent literature DOI: 10.3390/cancers17111784) by providing a solid context to highlight the relevance of radiomics as a non-invasive investigative tool.
We added the paragraph:
Melanoma is an aggressive skin cancer, driven by both environmental and genetic factors. Ultraviolet radiation induces DNA damage and mutations, contributing to genomic instability, while Infrared radiation, although non-mutagenic, may promote melanoma by inhibiting apoptosis in damaged melanocytes. Among the main genes implicated in genetic susceptibility are MC1R, CDKN2A, TP53, and components of the MAPK pathway, which regulate pigmentation, DNA repair, and cell proliferation. (https://doi.org/10.3390/cancers17111784). [...]
In the context of this malignancy, radiomics represents a non-invasive, reproducible diagnostic tool that may help to predict responses to therapy and survival outcomes, provide prognostic significance, assess tumor heterogeneity, detect mutational status, even evaluate the presence of tumor-infiltrating lymphocytes (TILs), especially CD8+ T cells, supporting more personalized treatment approaches.
- Revise the final sentence on line 93. The concluding sentence of the introduction, "With the aim to study the current applications and the state of art of radiomics in melanoma, we carried out a systemic review and meta-analysis," is somewhat generic. Since your abstract specifically mentions the meta-analysis on therapy response prediction, the introduction should foreshadow this key aspect. A more specific reformulation would be: "With the aim of synthesizing the existing literature and quantitatively evaluating the effectiveness of radiomic models..., we conducted a systematic review and meta-analysis..."
We modified the sentence, as for your suggestion.
With the aim of synthesizing the existing literature and quantitatively evaluating the effectiveness of radiomic models in predicting therapy response in melanoma, we conducted a systematic review and meta-analysis.
- Improve Figure 2. The current graph showing the trend over time is not visually impactful. For a clearer and more intuitive representation of discrete counts per year, replacing the line graph with a bar chart (histogram) would more effectively illustrate the number of studies published annually.
Thanks for this comment. We took your suggestion and modified the figure, accordingly.
